# 2D-Supervised Monocular 3D Object Detection by Global-to-Local Reconstruction

## Abstract

With the rise of big models, the need for data has become increasingly crucial. However, costly manual annotations may hinder further advancements. In monocular 3D object detection, existing works have investigated weakly supervised algorithms with the help of additional *LiDAR* sensors to generate 3D pseudo labels, which cannot be applied to ordinary videos. In this paper, we propose a novel paradigm called $BA^2$-Det that utilizes global-to-local 3D reconstruction to supervise the monocular 3D object detector in a *purely 2D* manner. Specifically, we use scene-level global reconstruction with global bundle adjustment (BA) to recover 3D structures from monocular videos. Then we develop the DoubleClustering algorithm to obtain object clusters. By learning from the generated complete 3D pseudo boxes in global BA, GBA-Learner can predict 3D pseudo boxes for other occluded objects. Finally, we train an LBA-Learner with object-centric local BA to generalize the 3D pseudo labels to moving objects. Experiments conducted on the large-scale Waymo Open Dataset show that the performance of $BA^2$-Det is on par with the fully-supervised BA-Det trained with 10% videos, and even surpasses some pioneering fully-supervised methods. Besides, as a pretraining method, $BA^2$-Det can achieve 20% relative improvement on KITTI dataset. We also show the great potential of $BA^2$-Det for detecting open-set 3D objects in complex scenes. Anonymous project page: `https://ba2det.site`.

## 1 Introduction

3D object detection has gained increasing attention from researchers and has become a fundamental task in real-world perception. Thanks to the efforts of researchers, 3D object detection using LiDAR and RGBD input has been shown to be practical in both traffic scenes (Shi et al., 2020; Yin et al., 2021; Fan et al., 2023) and indoor scenes (Qi et al., 2019; Liu et al., 2021a). Autonomous vehicles and service robots increasingly rely on 3D object detection to perceive and understand their environment. In recent years, detecting 3D objects from images has emerged as a growing area of research. Due to the cheap cost of the camera sensor, monocular and multi-camera 3D object detection is assured of a proper place in 3D perception. From the aspect of performance, camera-only 3D object detector is also gradually catching up with LiDAR-based methods.

However, with heavy and expensive manual annotations, the development of camera-based 3D object detection methods is potentially limited. The existing work has explored weakly supervised (Zakharov et al., 2020; Peng et al., 2022c) algorithms with the help of LiDAR to unlock the potential of unlabeled images. However, the camera-only 3D object label generation method has not been investigated much. In this paper, we expect to explore camera-only 3D object detection without 3D annotations and the auxiliary LiDAR data. Considering that 2D object detection and segmentation are nearly free (Kirillov et al., 2023) to obtain, in this paper we assume that we can get the 2D object labels and simplify the problem as 2D supervised monocular 3D object detection.

In this paper, we propose a novel paradigm for 2D supervised monocular 3D object detection, called $BA^2$-Det, by 3D reconstruction from global (scene-level) to local (object-level) using twice bundle adjustment (BA) technique, as shown in Fig. 1. We face and address some core issues step-by-step in this process. The first issue is: *how to recover the 3D location of each object only from images.* Given a video taken by a moving camera, the whole scene can be recovered in 3D by solving global BA (Schonberger & Frahm, 2016). Furthermore, with 2D bounding boxes in each frame, we group

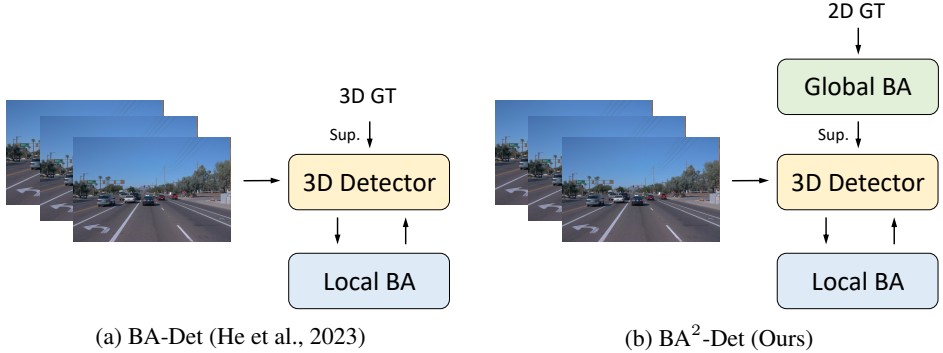

Figure 1: **A brief introduction of BA$^2$-Det.** Compared with BA-Det, to learn detector without 3D labels, our BA$^2$-Det generates 3D pseudo labels from scene-level reconstruction using global BA.

the reconstructed foreground 3D points into object clusters. Specifically, to obtain more complete and clean object clusters, we design a two-step clustering algorithm called *DoubleClustering*, including intra-frame Local Point Clustering (LPC) and inter-frame Global Point Clustering (GPC).

After obtaining 3D object clusters, the next problem is: *how to estimate the 3D bounding boxes from the object clusters.* A simple solution is to fit a tight 3D bounding box enclosing the object point cluster. However, the estimated orientation is very unreliable since it is sensitive to the outlier points in the cluster. We design a better orientation-optimized method by making sure more reconstructed points are near the edge of the 3D bounding box. Additionally, the bounding box may only encompass a portion of the object due to occlusion. So we learn the occluded object's complete 3D bounding box from other non-occluded 3D boxes with a PointNet-like (Qi et al., 2017) neural network, called *GBA-Learner*, i.e., learning from the global BA.

However, many moving objects are not reconstructed in scene-level global 3D reconstruction. So the last problem is: *how to generalize the 3D pseudo labels to dynamic objects.* Inspired by BA-Det (He et al., 2023), equipped with object-centric local BA, we design a monocular 3D object detector (*LBA-Learner*). We train LBA-Learner using iterative self-retraining from static objects. With the generalization ability of the object detector and object-centric local BA, we can predict the 3D bounding boxes for both static and moving objects.

In summary, our main contributions are as follows: (1) We propose a novel paradigm for 2D supervised monocular 3D object detection, called BA$^2$-Det, and integrate the generation of 3D pseudo labels and the learning process of the monocular 3D object detector from the perspective of global-to-local 3D reconstruction. (2) We aim to address three fundamental technical challenges in learning the 3D object detector without 3D labels. To achieve this, we have developed three main modules: DoubleClustering, GBA-Learner, and LBA-Learner. (3) We conducted experiments on various datasets, including the KITTI dataset, the large-scale Waymo Open Dataset (WOD), and open-set general scenes. The main results and ablation study show the high quality of pseudo labels. These pseudo labels can be used either for the direct training of 3D detectors or for leveraging large-scale data as pretraining to enhance the performance of fully supervised detection models. The performance of BA$^2$-Det is on par with the fully-supervised BA-Det trained with 10% videos and even outperforms some pioneering fully-supervised methods. As a pretraining method, BA$^2$-Det can bring 20% relative improvement on KITTI dataset. We also show the great potential for detecting open-set 3D objects in complex scenes.

## 2 RELATED WORK

### 2.1 MONOCULAR 3D OBJECT DETECTION

Monocular 3D object detection (Chen et al., 2016; Brazil & Liu, 2019; Wang et al., 2019; Zhang et al., 2021) has been explored for several years, thanks to the ability of the neural network that can estimate relatively accurate depth from the monocular images. The existing monocular 3D object detection methods can be divided into three categories: regressing 3D objects from the image directly, regressing on the depth map or lifted 3D space, and regressing based on geometric

constraints. CenterNet (Zhou et al., 2019) and FCOS3D (Wang et al., 2021c) are the representing works to estimate the 3D objects with a 3D regression branch based on the 2D object detectors CenterNet (Zhou et al., 2019) and FCOS (Tian et al., 2019), respectively. PL (Wang et al., 2019) and PL++ (You et al., 2020) use the off-the-shelf dense depth estimator to project the scene in 3D space and detect objects from pseudo-LiDAR. D4LCN (Ding et al., 2020) and PatchNet (Ma et al., 2020) use the image-aligned depth map to extract features. E2E-PL (Qian et al., 2020) and CaDDN (Reading et al., 2021) jointly learn the depth estimator and 3D object detector in an end-to-end manner. With the geometric constraints, the 3D object depth can be estimated by solving Perspective-n-Point (PnP) problem. MonoFlex (Zhang et al., 2021) solves PnP from the vertical lines of 3D bounding boxes. AutoShape (Liu et al., 2021b) uses semantic points sampled on the CAD model. DCD (Li et al., 2022b) uses arbitrary point pairs to construct dense constraints. Recently, there has been a surge in the development of temporal 3D object detection from monocular images. DfM (Wang et al., 2022a) and BA-Det (He et al., 2023) aggregate temporal information at the scene level and in an object-centric manner, inspired by two-view and multi-view geometry theory, respectively.

## 2.2 DETECTING 3D OBJECTS WITHOUT 3D LABELS

Since the 3D object detection task makes great progress in the past few years, many researchers begin to explore using fewer 3D labels or even without 3D labels to train a 3D object detector. For LiDAR-based 3D object segmentation, clustering-based methods (Triebel et al., 2010; Campello et al., 2013; Nunes et al., 2022) are the mainstream methods. LSMOL (Wang et al., 2022b) and Najibi et al. (Najibi et al., 2022) combine image and LiDAR to segment 2D and 3D objects. However, only class-agnostic segmentation can be achieved in these methods. MODEST (You et al., 2022) is an unsupervised 3D mobile object detection method to predict 3D bounding box. Its key idea is that mobile objects are ephemeral members of a scene. For image-based 3D object detection, SDFLabel (Zakharov et al., 2020) is a pioneer work that can auto-label the 3D bounding boxes from a pre-trained 2D detector and the corresponding LiDAR data by recovering the object shape with signed distance fields (SDF). WeakM3D (Peng et al., 2022c) is also a weakly supervised method and needs additional LiDAR data. Yang et al. (Yang et al., 2022) first explore the image-only weakly supervised without LiDAR. However, box size and orientation cannot be estimated in this method and can only be learned by ground truth in the semi-supervised setting. Unlike the above works, our BA$^2$-Det *only uses images* without LiDAR as an auxiliary modality and can estimate 3D *bounding boxes* including center position, box size, and orientation.

## 3 PRELIMINARY: BA-DET

BA-Det (He et al., 2023) is a two-stage 3D object detector with a learnable object-centric global optimization network. The BA-Det pipeline has two stages. The first stage is a single-frame monocular 3D object detector. The second stage is an object-centric temporal correspondence learning (OTCL) module. This module uses feature-metric object bundle adjustment loss to learn temporal feature correspondence. During inference, BA-Det uses object-centric bundle adjustment (OBA) to optimize the object's pose and 3D box size over time. It takes the first-stage object prediction and temporal feature correspondence as input.

We notice that for the moving objects, 3D pseudo labels cannot be generated in the scene-level global reconstruction. However, because BA-Det can handle both static and moving objects in an object-centric style, we can design a training strategy that solves this problem. We first learn the 3D object iteratively to refine the 3D bounding box using the first stage of BA-Det, and with the help of object-centric bundle adjustment, the moving objects that do not learn well can be refined in the second stage of BA-Det.

## 4 METHODOLOGY

**Problem setup.** In this paper, we present our BA$^2$-Det for detecting 3D objects from monocular images. The objects are represented as 7-DoF 3D bounding boxes with only yaw rotation considered. Our detector does not require training with 3D ground truth or the use of LiDAR data as an auxiliary modality. However, we use additional 2D labels to separate instances and obtain class labels for each object. Note that during inference, we do not use any types of labels.

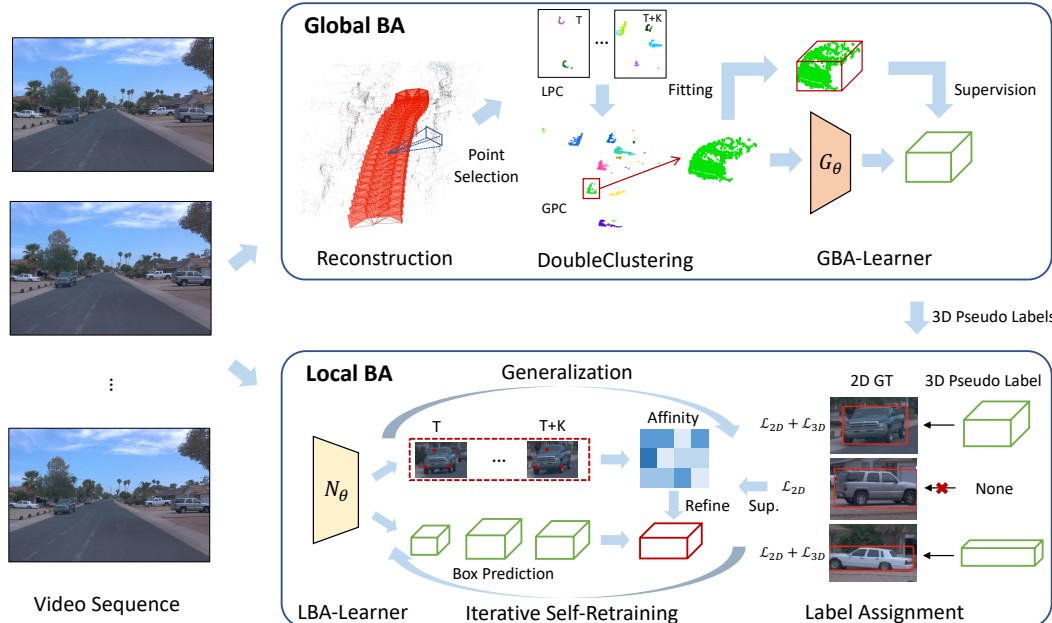

Figure 2: **Pipeline of BA$^2$-Det.** We take the video sequence as input. The Global BA stage is to generate 3D pseudo labels from scene-level global reconstruction, including DoubleClustering and GBA-Learner. Then Local BA is to learn a monocular 3D object detector in an iterative way.

**Algorithm overview.** We introduce our framework BA$^2$-Det briefly and explain module designs. In general, BA$^2$-Det is a pipeline learning reconstructed 3D objects from scene-level global reconstruction to object-centric local reconstruction. As shown in Fig. 2, BA$^2$-Det mainly contains three modules, DoubleClustering for obtaining 3D object clusters from scene reconstruction (Sec. 4.1), GBA-Learner learning 3D object bounding boxes from clusters (Sec. 4.2), and LBA-Learner learning a 3D object detector iteratively based on object-centric local reconstruction (Sec. 4.3).

## 4.1 DOUBLECLUSTERING: 3D OBJECT CLUSTERING FROM SCENE-LEVEL RECONSTRUCTION

Using the Structure-from-Motion (SfM) technique, it is possible to reconstruct a 3D scene from ego-motion. Then from the reconstructed scene, with the help of 2D bounding boxes in each frame, the 3D object cluster can be obtained by clustering the foreground points from the reconstructed scene. So, in this section, we introduce an algorithm called DoubleClustering (Alg. 1) for extracting 3D object clusters from the 3D reconstructed scene.

Firstly, let's revisit scene reconstruction with SfM. We denote the video sequence as $\mathcal{V} = \{\mathbf{I}_t | t = 1, 2 \cdots, T\}$, keypoints as $\mathbf{p}_t^i = [u_i, v_i]^\top, (i = 1, 2, \cdots, n)$ and local feature as $\mathcal{F}_t = \{\mathbf{f}_t^i\}$. In this paper, the keypoint extraction and local feature matching network are based on SuperPoint (DeTone et al., 2018) and SuperGlue (Sarlin et al., 2020). Given camera intrinsic parameter $\mathbf{K}$ and the camera extrinsic parameter $\mathbf{T}_t = [\mathbf{R}_t | \mathbf{t}_t]$ in $t$-th frame, 3D keypoint $\mathbf{P}_i$ in the global frame can be optimized by solving bundle adjustment with projection error

$$\{\mathbf{P}_i^*\}_{i=1}^n = \underset{\{\mathbf{P}_i\}_{i=1}^n}{\arg\min} \frac{1}{2} \sum_{i=1}^n \sum_{t=1}^T ||\mathbf{p}_i^t - \Pi(\mathbf{T}_t, \mathbf{P}_i, \mathbf{K})||^2, \tag{1}$$

where $\Pi(\cdot)$ is the function projecting the 3D points in the world frame to the image. However, when the ego moves slowly, the disparity is small and the observation noise can affect reconstruction. Therefore, when the camera moves slowly, we disregard the video sequence and do not reconstruct the scene.

After the scene reconstruction, we choose the 3D points that can be projected in the 2D bounding boxes, and perform the Local Point Clustering (LPC) in each frame via the Connected Component (CC) algorithm to choose the largest cluster for each object $\mathbf{B}_j^t$:

$$\mathcal{P}_j^t = \{\mathbf{P}_i\}_{i=1}^m = \text{LPC}(\{\mathbf{P}_i^* | \Pi(\mathbf{T}_t, \mathbf{P}_i^*, \mathbf{K}) \in \mathbf{b}_j^t\}), \tag{2}$$

---

**Algorithm 1** Generating 3D object clusters (DoubleClustering)

---

**Input:** video clip $\mathcal{V}$, camera intrinsic $\{\mathbf{K}\}$, camera pose $\{\mathbf{T}_t\}$, 2D bounding box with id $\{\mathbf{B}_j^t\}_{j=1}^{n_d}$
**Output:** 3D object cluster $\{\mathcal{P}_j^*\}_{j=1}^{n_d}$
1: $\{\mathbf{P}_i^*\}_{i=1}^n \leftarrow \text{SfM}(\mathcal{V}, \{\mathbf{K}\}, \{\mathbf{T}_t\})$;
2: $\mathcal{P} \leftarrow \varnothing$;
3: **for** $j \in [1, n_d]$ **do**                                    ▷ LPC for each object in each frame
4:     **for** $t \in [1, T]$ **do**
5:         $\mathcal{P}_j^t \leftarrow \text{LPC}(\{\mathbf{P}_i^*\}, \mathbf{B}_j^t)$;
6:         $\mathcal{P} \leftarrow \mathcal{P} \cup \mathcal{P}_j^t$;
7: $\{\mathcal{P}_{j'}\}_{j'=1}^{n_d'} \leftarrow \text{GPC}(\mathcal{P})$;                        ▷ GPC from $\mathcal{P}$ to generate $n_d'$ clusters
8: $\{\mathcal{P}_j^*\}_{j=1}^{n_d} \leftarrow \text{Match}(\{\mathcal{P}_{j'}\}, \{\mathbf{B}_j^t\})$;                   ▷ Match objects to the point clusters
9: **return** $\{\mathcal{P}_j^*\}_{j=1}^{n_d}$

---

where we denote $\mathbf{b}_j^t$ as the region in the 2D box $\mathbf{B}_j^t$. The distance threshold in CC algorithm is $\delta_1$.

Besides, we cluster the 3D points in the global scene by CC with distance threshold $\delta_2$, called Global Point Clustering (GPC):

$$\{\mathcal{P}_{j'}\}_{j'=1}^{n_d'} = \text{GPC}(\bigcup_{t,j} \mathcal{P}_j^t), \tag{3}$$

where $n_d'$ is the total object clusters. We ignore clusters with point numbers lower than threshold $\theta$. Finally, we choose the object cluster with the highest number of projected points in it as the corresponding cluster $\mathcal{P}_j^*$ for the 2D bounding box $\mathbf{B}_j^t$.

## 4.2 GBA-Learner: Generating 3D Object Labels from 3D Object Clusters

We now obtain object clusters and then we need to generate the 3D pseudo box from the object points in each cluster. To further learn and refine 3D object labels, we design a PointNet-based network called GBA-Learner. Given the object cluster $\mathcal{P}_j^*$ in the global frame, we first fit a tight 3D bounding box. A straightforward solution would be to find the minimum enclosing rectangle in the bird's-eye view (BEV) and calculate the box's height using points along the z-axis. However, this method is not effective enough in estimating orientation due to its susceptibility to noise points. Utilizing the assumption that the reconstructed points are mainly located on the object's surface and taking inspiration from Zhang et al. (Zhang et al., 2017), we optimize the orientation $r_y$ by minimizing the total distance between the points and their closest edge. Subsequently, we adjust the width and length of the bird's-eye view bounding box to achieve minimal area:

$$r_y^*, \mathbf{B}_{bev}^* = \underset{r_y \in [0,\pi), \mathbf{B}_{bev} \in \mathbb{R}^2}{\arg\min} \sum_{i=1}^m \min_{l \in [1,4]} d(\mathbf{P}_i, \mathbf{R}(r_y)\mathbf{B}_{bev}^l), \tag{4}$$

where $\mathbf{P}_i$ is the 3D points in the object cluster, and $\mathbf{R}(r_y)\mathbf{B}_{bev}^l$ is the edge of rotated BEV bounding box with angle $r_y$, we use $l_2$ distance as distance function $d(\cdot)$. The results in Table 3 and Table 14 also validate the effectiveness of our box fitting method, especially on the orientation metrics.

However, not all fitted 3D boxes are satisfactory because some objects may be occluded or affected by outliers, resulting in sizes that differ from the ground truth. Therefore, we design a neural network $G_\theta$ to learn and refine the initial 3D bounding box from well-reconstructed objects. This network takes the object cluster as input and normalizes the coordinates of 3D points using the center of the 3D pseudo box. It consists of a PointNet backbone and a head to predict 7DoF 3D bounding box $[c_x, c_y, c_z, w, h, l, r_y]$. To simulate the occlusion-induced partial object points, we perform data augmentation by cutting off regions from the well-reconstructed object. Note that we only consider the length between $[\sigma_0, \sigma_1]$ as the well-reconstructed object and take these 3D pseudo boxes to supervise $G_\theta$. The more details in training $G_\theta$ is in Sec. B in Appendix.

Note that we can only generate 3D object labels for the static object with $G_\theta$ because the moving objects cannot be reconstructed in the scene-level global reconstruction. So we still need a neural network to generalize to the moving objects by learning from static objects and to refine the 3D bounding boxes with object-centric local reconstruction.

### 4.3 LBA-Learner: 3D Object Detection with Object-centric Reconstruction

As the size and orientation-optimized 3D pseudo labels are generated, in this section, we train a 3D object detector called LBA-Learner. By using the object-centric LBA-Learner, we can achieve better generalization for moving objects and improve object temporal consistency. LBA-Learner $N_\theta$ is based on BA-Det (He et al., 2023). The detector learning contains *initial training* and *iterative self-retraining*.

The initial training stage is to learn a 3D object detector with generated 3D pseudo labels, which is different from fully supervised methods and the iterative self-retraining stage. There are two main differences: the distance distribution of the labels and some unlabeled 3D objects (usually moving objects). Firstly, 3D pseudo labels have a wider distance distribution due to the ability of scene-level global reconstruction to recover 3D object points from an entire video. However, the ground truth labeled on LiDAR is usually near due to the limited scanning range. We discuss this problem with experiments in Sec. 5.5 (Table 6). Secondly, regarding the issue of unlabeled 3D objects, we assign labels using 2D ground truth (GT) labels and disregard their 3D losses if there are no 3D pseudo labels. This ensures that the unlabeled objects are not considered negative samples. This strategy will keep the recall and enhance the refinement of the 3D position during the iterative self-retraining stage. The verification experiments are in Sec. 5.4 (Table 3).

Besides, another significant distinction between our BA²-Det and fully-supervised monocular 3D object detector is how to supervise the orientation estimation. The orientation of the 3D pseudo label may have a deviation of $180°$, as it can be difficult to distinguish whether the object is facing forward or backward. MultiBin Loss (Mousavian et al., 2017) is a common practice in fully supervised methods. We use it with modification to alleviate this problem. We calculate two losses using the original pseudo label and the $180°$-reversed one, selecting the minimum as the final orientation loss. Experiment results are in Table 14. As for the other losses, we mainly follow BA-Det (He et al., 2023).

Because the 3D pseudo labels are not precise enough and the camera-based 3D object detector is hard to learn, we iteratively self-retrain the detector with predictions as the updated pseudo labels. We design two retraining strategies: keeping the initial 3D pseudo labels and supplementing the high score predictions iteratively

$$\mathcal{D}^{(l)}(X,Y) = N_\theta^{(l)}(\widetilde{\mathcal{D}}(X,Y) \cup \mathcal{D}^{(l-1)}(X,Y)),$$ 
(5)

or using 3D pseudo labels for initial training and updating labels with predictions from last iteration

$$\begin{cases} \mathcal{D}^{(0)}(X,Y) = N_\theta^{(0)}(\widetilde{\mathcal{D}}(X,Y)), \\ \mathcal{D}^{(l)}(X,Y) = N_\theta^{(l)}(\mathcal{D}^{(l-1)}(X,Y)), \end{cases}$$
(6)

where $\widetilde{\mathcal{D}}(X,Y)$ is the dataset with generated pseudo labels, $\mathcal{D}^{(l)}(X,Y)$ is the dataset with predicted labels, $(l)$ means the $l$-th self-training iteration. Note that we do not keep the last network parameters for each self-retraining iteration and train the network $N_\theta$ for the same $\kappa$ epochs. We finally choose the second retraining strategy for better performance (Table 5 in Sec. 5.5).

## 5 Experiments

### 5.1 Datasets and metrics

**Waymo open dataset (WOD).**  To verify our proposed BA²-Det, we conduct our ablation studies and comparison experiments with other methods on the large-scale autonomous driving dataset, Waymo Open Dataset (WOD) (Sun et al., 2020). WOD is the mainstream 3D object detection benchmark, containing 1150 video sequences, 798 for training, 202 for validation, and 150 for testing. Only objects within 75m that can be scanned by LiDAR have 3D labels. To keep the same experiment settings as other methods, we mainly report the results on WOD v1.2. The evaluation metrics for camera-based 3D object detection are 3D AP and LET-3D AP (Hung et al., 2022). 3D AP is a common metric for both camera and LiDAR-based 3D object detection. LET-3D AP is specifically designed for camera-only 3D object detection. Because the camera-based 3D object detector has a natural weakness in depth estimation, LET-3D AP is much looser for longitudinal localization

Table 1: **The main results on WOD *val* set.** '3D Sup.' means the ratio of video sequences with 3D labels. †: trained with 1/3 frames. *: without object-centric BA refinement.

| Method | 3D Sup. | 3D AP$_5$ | 3D APH$_5$ | 3D AP$_{50}$ | 3D APH$_{50}$ | LET APL$_{50}$ | LET AP$_{50}$ | LET APH$_{50}$ |
|---|---|---|---|---|---|---|---|---|
| PatchNet (Ma et al., 2020) | 100%† | - | - | 2.92 | 2.74 | - | - | - |
| M3D-RPN (Brazil & Liu, 2019) | 100%† | - | - | 3.79 | 3.63 | - | - | - |
| PCT (Wang et al., 2021a) | 100%† | - | - | 4.20 | 4.15 | - | - | - |
| MonoJSG (Lian et al., 2022) | 100%† | - | - | 5.65 | 5.47 | - | - | - |
| GUPNet (Lu et al., 2021) | 100%† | - | - | 10.02 | 9.94 | - | - | - |
| MonoFlex (Zhang et al., 2021) | 100% | 70.33 | 69.41 | 34.70 | 34.43 | 50.63 | 67.30 | 66.50 |
| BA-Det (He et al., 2023) | 100% | 72.96 | 71.78 | 40.93 | 40.51 | 54.45 | 68.32 | 67.36 |
| MonoFlex (Zhang et al., 2021) | 10% | 53.68 | 52.30 | 15.44 | 15.22 | 28.21 | 44.21 | 43.23 |
| BA-Det (He et al., 2023) | 10% | 57.29 | 55.27 | 19.70 | 19.27 | 32.53 | 46.91 | 45.52 |
| SfM+BA-Det (Baseline) | 0% | 27.84 | 8.80 | 2.89 | 0.75 | 7.34 | 10.75 | 3.31 |
| **BA$^2$-Det* (Ours)** | 0%† | 55.24 | 40.87 | 6.24 | 5.37 | 16.61 | 27.94 | 21.32 |
| **BA$^2$-Det* (Ours)** | 0% | 56.33 | 42.05 | 6.97 | 6.00 | 17.87 | 29.62 | 22.8 |
| **BA$^2$-Det (Ours)** | 0% | **60.01** | **44.81** | **10.39** | **8.98** | **22.24** | **32.60** | **23.86** |

and uses Longitudinal Error Tolerant IoU (LET-IoU) instead of the original IoU as the criterion. Following the existing camera-based 3D object detection methods, we mainly report the results of the VEHICLE class on the FRONT camera. For 3D AP and 3D APH, we choose a loose IoU threshold of 0.05 and a common one of 0.5, called AP$_5$ and AP$_{50}$. For LET-3D metrics, we report the results under the official IoU threshold of 0.5.

**KITTI dataset.** KITTI object detection benchmark consists of 7481 images for training and 7518 images for testing. Unlike WOD, it is not organized as long video sequences. The main evaluation metric is 3D AP on three difficulty levels, easy, moderate, and hard. Besides the object detection benchmark, KITTI also provides the raw dataset without 3D object labels. We train BA$^2$-Det on KITTI raw dataset.

## 5.2 IMPLEMENTATION DETAILS

**Architecture.** The main 3D object detector architecture we used follows BA-Det (He et al., 2023). We use a DLA-34 (Yu et al., 2018) as the backbone without an FPN neck, and the head is with 2 layers of 3×3 convolutions and MLP. The resolution of the input images is 1920×1280. If the input size is smaller than it, we will use zero padding to complete the image.

**Label generation and model training.** The scene reconstruction is based on hloc (Sarlin et al., 2019) framework[1]. The distance threshold $\delta_1$ and $\delta_2$ in DoubleClustering are 0.5 and 0.7. We keep the object cluster for more than $\theta = 100$ points. The size threshold $\sigma_0 = 3m$ and $\sigma_1 = 10m$. Our implementation is based on the PyTorch (Paszke et al., 2019) framework. We train our model on 8 NVIDIA RTX 4090 GPUs. Adam (Kingma & Ba, 2014) optimizer is applied with $\beta_1 = 0.9$ and $\beta_2 = 0.999$. The learning rate is $8 \times 10^{-5}$ and weight decay is $10^{-5}$. We train 1 epoch for network $G_\theta$ and $\kappa = 12$ epochs for the 3D object detector $N_\theta$. The loss weights are the same as BA-Det. The self-retrain iteration number for $N_\theta$ is 2. As shown in Table 6, we choose labels in the wider depth range [0.5m, 200m] for initial training and regular range [0.5m, 75m] for iterative self-retraining.

## 5.3 MAIN RESULTS

As shown in Table 1, we show the main results compared with fully-supervised BA-Det, a simple solution combining SfM and BA-Det, and some other fully supervised methods. Especially, to make a clear understanding of our results, we also compare the results of BA-Det trained with fewer data. We find that with a loose IoU threshold (0.05), our BA$^2$-Det can outperform fully-supervised BA-Det with 10% data by 2.7 AP and is close to the 100% data (with a ~12 AP gap). As for the 0.5 IoU threshold, we can beat some other fully-supervised methods, such as PCT and MonoJSG. Compared with our baseline method, using SfM (Schonberger & Frahm, 2016) and clustering to fit 3D labels and learning a BA-Det, our BA$^2$-Det have huge gains on all metrics.

We also conduct additional experiments on KITTI to compare with other SOTA methods. We report 3D object detection results on test set. Note that (1) there is no long video for 3D reconstruction on

---

[1]https://github.com/cvg/Hierarchical-Localization

Table 2: **The results on *test* set of KITTI detection benchmark.** The main evaluation metric is 3D $AP_{IoU=0.7|R_{40}}$ on three difficulty levels, easy, moderate, and hard.

| Method | Reference | Extra Data | Easy | Moderate | Hard |
|---|---|---|---|---|---|
| PatchNet (Ma et al., 2020) | ECCV 2020 | KITTI Raw image+depth | 15.68 | 11.12 | 10.17 |
| MonoDTR (Huang et al., 2022) | CVPR 2022 | - | 21.99 | 15.39 | 12.73 |
| DCD (Li et al., 2022a) | ECCV 2022 | CAD models | 23.81 | 15.90 | 13.21 |
| MonoJSG (Lian et al., 2022) | CVPR 2022 | - | 24.69 | 16.14 | 13.64 |
| DID-M3D (Peng et al., 2022b) | ECCV 2022 | - | 24.40 | 16.29 | 13.75 |
| MonoDETR (Zhang et al., 2023) | ICCV 2023 | - | 25.00 | 16.47 | 13.58 |
| MonoATT (Zhou et al., 2023) | CVPR 2023 | - | 24.72 | 17.37 | 15.00 |
| MonoNeRD (Xu et al., 2023) | ICCV 2023 | - | 22.75 | 17.13 | 15.63 |
| LPCG-Monoflex (Peng et al., 2022a) | ECCV 2022 | KITTI Raw image+LiDAR | 25.56 | 17.80 | 15.38 |
| CMKD (Hong et al., 2022) | ECCV 2022 | KITTI Raw image+LiDAR | 28.55 | 18.69 | 16.77 |
| MonoXiver (Liu et al., 2023) | ICCV 2023 | - | 25.24 | 19.04 | 16.39 |
| MonoFlex (Zhang et al., 2021) | CVPR 2021 | - | 19.94 | 13.89 | 12.07 |
| **BA$^2$-Det+MonoFlex (Ours)** | - | KITTI Raw image | 23.45 | 16.30 (+2.41) | 13.50 |

Table 3: **The ablation study about each component of BA$^2$-Det.**

| $N_\theta$ w/ 3D | $N_\theta$ w/ 2D | $G_\theta$ | $r_y$ w/ $d$ | Iter. | OBA | 3D AP$_5$ | 3D APH$_5$ | LET APL$_{50}$ | LET AP$_{50}$ |
|---|---|---|---|---|---|---|---|---|---|
| ✓ | | | | | | 20.97 | 6.70 | 4.27 | 7.28 |
| | ✓ | | | | | 28.40 | 11.34 | 5.02 | 8.62 |
| | ✓ | ✓ | | | | 33.75 | 11.94 | 9.63 | 16.80 |
| | ✓ | ✓ | ✓ | | | 41.17 | 28.73 | 12.23 | 21.41 |
| | ✓ | ✓ | ✓ | ✓ | | 56.33 | 42.05 | 17.87 | 29.62 |
| | ✓ | ✓ | ✓ | ✓ | ✓ | **60.01** | **44.81** | **22.24** | **32.60** |

KITTI object detection dataset, and we have to generate 3D pseudo boxes on KITTI raw dataset; (2) there are hardly any comparable methods available for 2D supervised 3D detection. So, we report the results taking BA$^2$-Det as the pretraining approach. We pretrain BA$^2$-Det on KITTI raw dataset without any labels (We use the 2D object detector Mask R-CNN trained on COCO) and finetune the monocular 3D object detector MonoFlex with 3D ground truth on KITTI detection training set. We only use a single frame during inference for a fair comparison. The results have been shown in Table 2. BA$^2$-Det can have 2.4 AP gain (about 20% relative improvement) on the moderate level.

## 5.4 ABLATION STUDY

We conduct the ablation study on WOD val set. The results are shown in Table 3. '$N_\theta$ w/ 3D' and '$N_\theta$ w/ 2D' mean we assign labels based on 3D or 2D labels (Sec. 4.3). We finally choose to assign labels with 2D labels and ignore the 3D loss if the 3D labels cannot be generated, which is 7.4 AP higher than only assigning labels when the 3D label exists. $G_\theta$ is the GBA-Learner, learning a complete 3D box from the partially reconstructed object, that can obtain a 5.4 AP gain. '$r_y$ w/ $d$' is the orientation optimization method minimizing the distances from points to the edge (Sec. 4.2), improving 16.8 APH by refining the orientation and box size. iterative self-retraining (Iter.) has the greatest gain of 15.2 AP. More experiments and discussions about iterative self-retraining are in Sec. 5.5. With object-centric bundle adjustment (OBA), the 3D bounding boxes are refined in temporal, which improves 3.7 AP.

## 5.5 DISCUSSIONS

**Self-retraining iteration.** We discuss the number of iterations required for the self-retraining stage. In Table 4, we conduct the experiments of a maximum of 4 iterations. The first iteration can bring the greatest benefit of 10.7 AP and the benefits are progressively decreased. After the second iteration, the performance is near the highest performance, which shows fast convergence of the self-retraining stage.

Table 4: **Discussion about self-retraining iterations.**

| Iter. | 3D AP$_5$ | 3D APH$_5$ | LET APL$_{50}$ | LET AP$_{50}$ |
|---|---|---|---|---|
| 0 | 41.17 | 28.73 | 12.23 | 21.41 |
| 1 | 51.81 | 38.66 | 17.42 | 28.57 |
| 2 | 56.33 | 42.05 | **17.87** | **29.62** |
| 3 | 56.70 | 42.27 | 16.87 | 28.06 |
| 4 | **57.12** | **42.42** | 16.64 | 27.73 |

**Retraining strategy.** In Sec. 4.3, we propose two self-retraining strategies (Eq. 5 and Eq. 6). In Table 5, we show the experiment results to retain 1 iteration. The latter is better than the former strategy. The combination of pseudo-labels and predictions may introduce two inconsistent data distri-

Table 5: **Ablation study of self-retraining strategy.**

| Strategy | 3D AP$_5$ | 3D APH$_5$ | LET APL$_{50}$ | LET AP$_{50}$ |
|---|---|---|---|---|
| w/o iter. | 41.17 | 28.73 | 12.23 | 21.41 |
| Eq. 5 | 39.71↓ 1.5 | 29.79↑ 1.0 | 12.75↑ 0.5 | 21.08↓ 0.3 |
| Eq. 6 | 51.81↑ 10.6 | 38.66↑ 9.9 | 17.42↑ 5.2 | 28.57↑ 7.2 |

butions, causing poor results. The results in Table 3 and Table 4 also indicate the importance of the chosen self-retraining strategy.

**Depth threshold settings.** By utilizing reconstruction, we can generate labels for objects that are farther away, and thus the depth distribution is different from ground truth, shown in Fig. 3. According to experiments (Table 6), we find that when we initially train the object detector, we

Table 6: **Experiments on different depth thresholds.**

| Iter. | Depth (m) | 3D AP$_5$ | 3D APH$_5$ | LET APL$_{50}$ | LET AP$_{50}$ |
|---|---|---|---|---|---|
| 0 | $[0.5, 75]$ | 22.24 | 7.66 | 4.83 | 8.56 |
| 0 | all, $[0.5, 200]$ | **33.75** | **11.94** | **9.63** | **16.80** |
| 1 | $[0.5, 75]$ | **42.07** | **15.00** | **12.32** | **20.48** |
| 1 | all, $[0.5, 200]$ | 40.60 | 14.49 | 12.18 | 20.05 |

need these farther objects, and for iteratively self-retraining, we can only train with the same depth ranges as the ground truth (0-75m).

Table 7: **Genralization ability of 3D localization on WOD training set.** 'w/o p.l.': objects without pseudo label, 'w/ p.l.': objects with pseudo label.

|  | Ratio | $\delta < 1.25 \uparrow$ | $\delta < 1.25^2 \uparrow$ | $\delta < 1.25^3 \uparrow$ | Abs Rel↓ | Sq Rel↓ | RMSE↓ | RMSE log↓ |
|---|---|---|---|---|---|---|---|---|
| w/o p.l. | 52.2% | 0.991 | 0.994 | 0.995 | 0.055 | 0.324 | 2.773 | 0.100 |
| w/ p.l. | 47.8% | 0.995 | 0.996 | 0.997 | 0.049 | 0.117 | 1.726 | 0.077 |
| All | 100% | 0.992 | 0.995 | 0.996 | 0.053 | 0.266 | 2.524 | 0.094 |

**Generalization to unlabeled objects.** In Table 7, we validate the performance of generalizing 3D location for unlabeled objects. The metrics follow depth estimation but on the object level. We find that only 47.8% objects can be generated 3D pseudo labels directly. The others are either heavily occluded or moving. The small gap between labeled and unlabeled objects in performance shows the generalization of LBA-Learner module to unlabeled moving objects.

**Open-set 3D object detection.** We demonstrate the capability to detect open-set 3D objects in complex scenes using SAM (Kirillov et al., 2023) instead of relying on 2D ground truth (Fig. 6 in appendix).

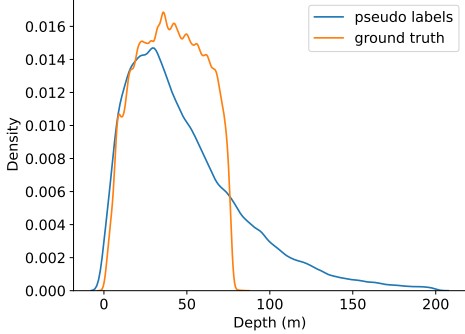

Figure 3: **Depth distributions of ground truth and pseudo labels.**

**3D Tracking results, more ablation studies, discussions, and qualitative results.** Please refer to Section B and Section C in the appendix. We show the qualitative of 3D object detection and tracking in Figure 7 in the appendix. For video demos, please refer to `https://ba2det.site`.

## 6 CONCLUSION

In this paper, we propose BA$^2$-Det, a novel paradigm for 2D supervised monocular 3D object detection. The key idea of BA$^2$-Det is to generate 3D pseudo labels and learn a 3D object detector from scene-level global reconstruction and object-centric local reconstruction. Specifically, the pipeline of BA$^2$-Det contains three parts: DoubleClustering algorithm to cluster object clusters from the reconstructed 3D scene, 3D object label generation with GBA-Learner, and 3D object detector LBA-Learner generalizing the pseudo labels to unlabeled dynamic objects. Experiments on the large-scale Waymo Open Dataset show that the performance of BA$^2$-Det is on par with the fully-supervised BA-Det trained with 10% videos and even outperforms some pioneering fully-supervised methods. As a pretraining method, BA$^2$-Det can bring 20% relative improvement on KITTI dataset. We also show the potential of BA$^2$-Det for detecting open-set 3D objects in complex scenes.

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

# A APPENDIX

In the appendix, we provide additional quantitative and qualitative experiment results. Especially,

- In Sec. B, we provide more quantitative experiment results, including more ablation studies, discussions and 3D multiple object tracking (MOT) results.

- In Sec. C, we show some qualitative results about the 2D supervised monocular 3D object detection, open-set 3D object detection, and multiple object tracking.

- In Sec. E and Sec. F, we also discuss the limitation of the proposed $BA^2$-Det and future work, and reproducibility statements to better reproduce our work.

# B ADDITIONAL EXPERIMENTS

## B.1 3D TRACKING RESULTS

Table 8: **Comparisons with SOTA methods on WOD for 3D MOT.**

|  | Fully Sup. | $MOTA_{50}$ ↑ | $Mismatch_{50}$ ↓ | $MOTA_{30}$ ↑ | $Mismatch_{30}$ ↓ |
|---|---|---|---|---|---|
| QD-3DT (Hu et al., 2022) | ✓ | 0.0308 | 0.00550 | 0.1867 | 0.01340 |
| CC-3DT (Fischer et al., 2022) | ✓ | 0.0480 | 0.00180 | 0.2032 | 0.00690 |
| SfM+BA-Det+Immortal (Wang et al., 2021b) |  | 0.0011 | <**0.00001** | 0.0652 | 0.00038 |
| $BA^2$-Det (Ours) |  | **0.0352** | 0.00002 | **0.1522** | **0.00008** |

In Table 8, we show additional 3D MOT results on WOD. We compare the proposed $BA^2$-Det with other SOTA monocular 3D MOT methods on WOD. All results are reported in Vehicle LEVEL 2 difficulty. 50 and 30 in the metrics are for the IoU threshold of 0.5 and 0.3. Note that the SOTA methods are both fully supervised, learning with 3D ground truth. Our $BA^2$-Det is a 2D supervised 3D multiple object tracking method. Even though, $BA^2$-Det is also comparable to these fully supervised ones. Especially for $MOTA_{50}$ metric, we outperform QD-3DT (Hu et al., 2022). For Mismatch, we only have less than 1/100 identity switches compared with the SOTA methods. Compared with the baseline method, using SfM to generate pseudo labels, BA-Det as the 3D object detector, and ImmortalTracker to track 3D objects, our method $BA^2$-Det improves the performance by a large margin.

## B.2 ADDITIONAL ABLATION STUDIES AND DISCUSSIONS

**Detailed results in different depth ranges.** The detailed results in different depth ranges are shown in Table 9. Compared with the baseline, we have a more significant gain for the objects far off.

Table 9: **The detailed results in different depth ranges (meters) on WOD *val* set.**

| Method | 3D Sup. | 3D $AP_5$ | | | 3D $APH_5$ | | | LET $APL_{50}$ | | | LET $AP_{50}$ | | |
|---|---|---|---|---|---|---|---|---|---|---|---|---|---|
|  |  | 0-30 | 30-50 | 50-∞ | 0-30 | 30-50 | 50-∞ | 0-30 | 30-50 | 50-∞ | 0-30 | 30-50 | 50-∞ |
| BA-Det | 100% | 87.80 | 72.52 | 48.45 | 86.91 | 71.52 | 46.98 | 66.15 | 57.97 | 36.44 | 82.74 | 69.58 | 45.77 |
| BA-Det | 10% | 73.25 | 54.00 | 34.50 | 71.38 | 52.22 | 32.53 | 38.31 | 35.57 | 22.40 | 56.98 | 47.28 | 31.11 |
| SfM+BA-Det | 0% | 46.87 | 25.88 | 9.09 | 14.26 | 8.86 | 2.84 | 11.35 | 7.74 | 2.60 | 17.59 | 10.12 | 3.48 |
| $BA^2$-Det (Ours) | 0% | 77.38 | 54.95 | 33.74 | 64.54 | 37.57 | 21.64 | 25.00 | 23.97 | 14.63 | 39.24 | 31.73 | 20.30 |

**Robustness across various levels of reconstruction quality.** Due to variations in reconstruction quality being primarily caused by differences in descriptors and matching algorithms, we simulate a scenario where there is a 25% decrease in the number of matched points due to failed matching. The results are presented in Table 10. The results show that our method is robust to the worse reconstruction quality.

Table 10: **Results with worse reconstruction.** We simulate worse point matching case.

| | 3D AP$_5$ | 3D APH$_5$ | LET APL$_{50}$ | LET AP$_{50}$ |
|---|---|---|---|---|
| 75% points | 37.02 | 26.20 | 9.75 | 17.19 |
| 100% points | **41.17** | **28.73** | **12.23** | **21.41** |

**Robustness across various levels of 2D annotation quality.** We discuss two kinds of worse 2D annotation qualities. The first is the influence of box numbers. We randomly drop 5% 2D bounding boxes and add 5% false positives. The second is for the 2D bounding box position. We add a maximum of 20% position error to the 4 corner points of the 2D bounding box. The results are shown in Table 11. Our method is relatively robust to the 2D box quality. Note that 2D object detection is a very stable and reliable technology. The 2D object detector is usually no worse than this simulation experiment. (For vehicles, >80 AP under 0.7 IoU threshold.)

Table 11: **Results using worse 2D annotations.**

| | 3D AP$_5$ | 3D APH$_5$ | LET APL$_{50}$ | LET AP$_{50}$ |
|---|---|---|---|---|
| GT box | 41.17 | 28.73 | 12.23 | 21.41 |
| Add FP + FN | 36.96 | 26.24 | 9.34 | 16.51 |
| Inaccurate box position | 31.34 | 21.88 | 8.45 | 14.81 |

**Necessity of Local Point Clustering (LPC).** LPC has two main roles: (1) provide semantic labels (class and ID) for reconstructed 3D points; (2) remove background points in the 2D box (2D box is not tight enough for the object boundary). If only using GPC to remove background points, the background points in frame A may be close to the foreground points in frame B (because there are more points, points are near to each other globally), and thus the clustering is not easy. Besides, without LPC, there will be more points to be clustered in GPC, which takes more memory and time. We conduct an additional ablation study (Table 12), i.e., we compare with using all points in 2D bounding box for GPC. The experiment also shows the effectiveness of our design of DoubleClustering.

Table 12: **Ablation study on DoubleClustering algorithm.**

| | 3D AP$_5$ | 3D APH$_5$ | LET APL$_{50}$ | LET AP$_{50}$ |
|---|---|---|---|---|
| GPC | 32.21 | 22.30 | 8.24 | 14.97 |
| LPC+GPC | **41.17** | **28.73** | **12.23** | **21.41** |

**Learning process details for GBA-Learner.** As mentioned in Section 4.2, we use the length of pseudo 3D bounding boxes to determine whether the object is *well-reconstructed*. This is a very simple yet effective metric, because we only need to find a good 3D box instead of its 3D surface in the object detection task. As mentioned in Section 4.2, GBA-Learner can improve box size estimation accuracy. Here, we report size prediction error, as the average absolute relative error of length, width and height in Table 13.

Besides, in training GBA-Learner, we do not simulate too large objects because we set a very loose length threshold. A vehicle whose length is too big (>10m) is very rare. According to the statistical results of GT (Fig. 4), it shows that less than 1% of objects have a length greater than 10m. We suppose that when the pseudo label is longer than 10m, it is likely that multiple objects have been incorrectly clustered into one cluster, resulting in a false positive. So we ignore this kind of cluster instead of learning a 3D box from it. As for the small pseudo box, we think it is a partially reconstructed object, and we can learn a full 3D box for it.

**Orientation error.** Our designs of orientation optimization (Sec. 4.2) and orientation loss can help orientation esitmation. Besides APH metric, we also report more detailed orientation metrics. As

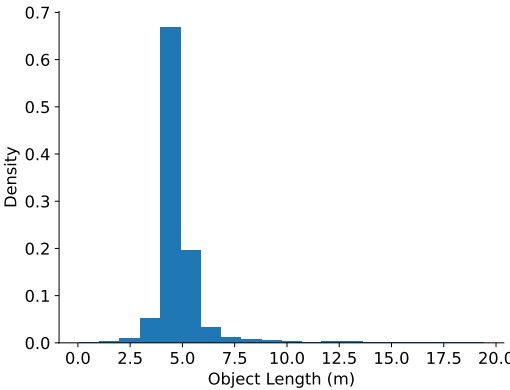

Figure 4: **Statistical results of GT object length distribution.**

Table 13: **Ablation study of GBA-Learner.** We also report results under the size prediction error metric.

| | 3D AP$_5$ | 3D APH$_5$ | LET APL$_{50}$ | LET AP$_{50}$ | Avg. Abs. Rel. |
|---|---|---|---|---|---|
| w/o $G_\theta$ | 28.40 | 11.34 | 5.02 | 8.62 | 0.076 |
| w/ $G_\theta$ | **33.75** | **11.94** | **9.63** | **16.80** | **0.063** |

for the orientation metrics, we report the average absolute relative error of orientation, defined as $\min(|\widetilde{\theta} - \theta|, 2\pi - |\widetilde{\theta} - \theta|)/\pi$, where $\widetilde{\theta}$ and $\theta$ are the predicted heading and the ground truth heading. In Table 14, we discuss the main factors to influence the orientation estimation, (1) minimizing the sum of distance ($r_y$ w/ d) and (2) orientation loss.

Table 14: **Impact of orientation optimization and loss design for orientation estimation.**

| | 3D AP$_5$ | 3D APH$_5$ | $r_y$ Abs. Rel. |
|---|---|---|---|
| minAreaRectangle + MultiBin loss | 33.75 | 11.94 | 0.205 |
| $r_y$ w/ d + MultiBin loss | 38.39 | 24.33 | 0.099 |
| $r_y$ w/ d + Orientation loss | **41.17** | **28.73** | **0.072** |

**Moving object filtering in pseudo label generation.** The points on moving objects are mostly ignored in SfM. To further alleviate the effect of these points, as mentioned in Sec. 4.1 and Sec. 5.2, we filter the object that has few points. We only keep the object cluster for more than $\theta = 100$ points. These objects may be dynamic objects that are not reconstructed well. The influence of this operation is ablated in Table 15.

**Pedestrian and cyclist categories.** Although we report the results of VEHICLE class in main paper, we would like to discuss the potential of BA$^2$-Det for other categories as well. Other categories such as pedestrians and bicycles are primarily dynamic objects that would be affected by the moving object filtering operation in global scene reconstruction.

- As for 3D pseudo-label generation, "the points on moving objects are mostly ignored". That means 3D pseudo labels are rarely generated for pedestrians and cyclists due to their movement. (However, when they are static, such as waiting for a red light, the pseudo label can still be generated.) This is the natural shortage of SfM.

- Although lacking 3D pseudo labels, we can utilize the "generalization ability of depth from other objects" of the monocular 3D object detector. The other objects are mainly vehicles that are learned with 3D pseudo labels. We can generalize the depth of pedestrians/cyclists from the learned depth of vehicles. This generalization ability is because (1) Depth is the class-agnostic attribute of the object,

Table 15: **Ablation study on few-point object filtering.**

|  | 3D AP$_5$ | 3D APH$_5$ | LET APL$_{50}$ | LET AP$_{50}$ |
|---|---|---|---|---|
| ALL | 36.03 | 24.92 | 9.27 | 16.38 |
| Filter <100 points | **41.17** | **28.73** | **12.23** | **21.41** |

Table 16: **Additional results on Pedestrian and Cyclist categories.** The middle columns show the object-level depth estimation results and the rightmost column shows the object-level orientation estimation results.

| Category | $\delta < 1.25 \uparrow$ | $\delta < 1.25^2 \uparrow$ | $\delta < 1.25^3 \uparrow$ | Abs Rel$\downarrow$ | Sq Rel$\downarrow$ | RMSE$\downarrow$ | RMSE log$\downarrow$ | $r_y$ Abs. Rel. |
|---|---|---|---|---|---|---|---|---|
| Car | 0.993 | 0.995 | 0.996 | 0.093 | 0.558 | 3.367 | 0.198 | 0.0805 |
| Pedestrian | 0.981 | 0.992 | 0.994 | 0.055 | 0.292 | 3.086 | 0.104 | 0.0804 |
| Cyclist | 0.848 | 0.950 | 0.962 | 0.120 | 0.804 | 4.491 | 0.168 | 0.3737 |

and the network learns depth from the whole image. During the inference stage, the network can leverage the learned depth information of other vehicles in the same image to predict the depth of pedestrians and cyclists. (2) We train monocular 3D object detector with 2D assignment strategy. That means we will predict depth for all 2D objects, both vehicles and pedestrians/cyclists.

We show the object-level depth and orientation estimation accuracy in Table 16. As we can see, the average depth accuracy for pedestrians is no worse than vehicles. The training samples for the cyclist are too rare, and thus slightly affect the performance of the cyclist.

## C QUALITATIVE RESULTS

We show some qualitative results about 3D object detection and tracking (BA$^2$-Det), open-set 3D object detection, and 2D MOT with auxiliary 3D representation (BA$^2$-Det). For more qualitative results and video demos, please refer to the project page: `https://ba2det.site`.

### C.1 3D OBJECT DETECTION AND MOT RESULTS ON WOD

In Fig. 7, we show the qualitative results of BA-Det (trained with 10% labeled videos), the baseline method, and the proposed BA$^2$-Det. Our method can achieve comparable performance with fully supervised BA-Det, and even better in some near cases. Compared with the baseline, a very obvious phenomenon is that our recall can be much better than the baseline method, mainly due to the iterative self-retraining design. The illustrations also show a typical failure case of BA$^2$-Det that on a distance of about 75m, there are some false positives. This is because the 3D pseudo labels can be 0-200m and thus somewhat affects the training process. If the annotations include some farther objects, this problem may be alleviated.

### C.2 OPEN-SET 3D OBJECT DETECTION WITH SAM

In Fig. 6, we also show the ability to detect open-set 3D objects in complex scenes with SAM (Kirillov et al., 2023) instead of 2D ground truth. We click the objects to generate the 2D masks in SAM. Please refer to the detailed video demos from `https://ba2det.site`.

## D DISCUSSION ABOUT NERF-FROM-IMAGE (PAVLLO ET AL., 2023)

We try nerf-from-image in our complex dataset WOD. However, only for object reconstruction, the performance is not as good as it shows in clean datasets CUB Birds and Pascal3D+ Cars. We analyze and discuss why it doesn't work on WOD in the following. In this section, we show the qualitative results.

- The main problem is for the small objects, i.e., for the objects far from the camera. Since nerf-from-image is designed for object-centric datasets, in which objects are near the camera

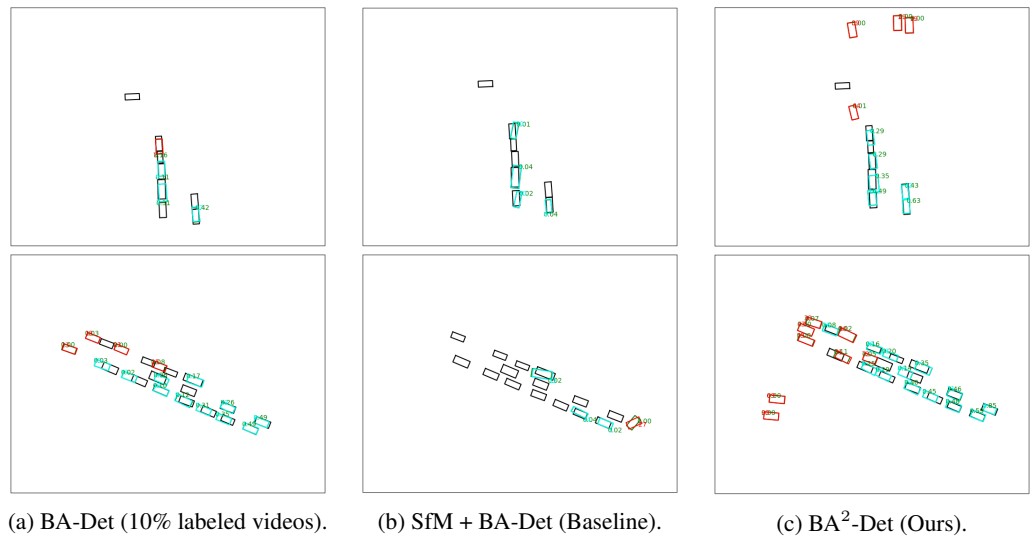

(a) BA-Det (10% labeled videos).  (b) SfM + BA-Det (Baseline).  (c) BA²-Det (Ours).

Figure 5: **Qualitative results of 3D object detection and tracking shown in BEV. Black** boxes are the ground truth, **cyan** boxes are the tracking results with id, **green** boxes are the detection results with scores, **red** boxes are the false positives.

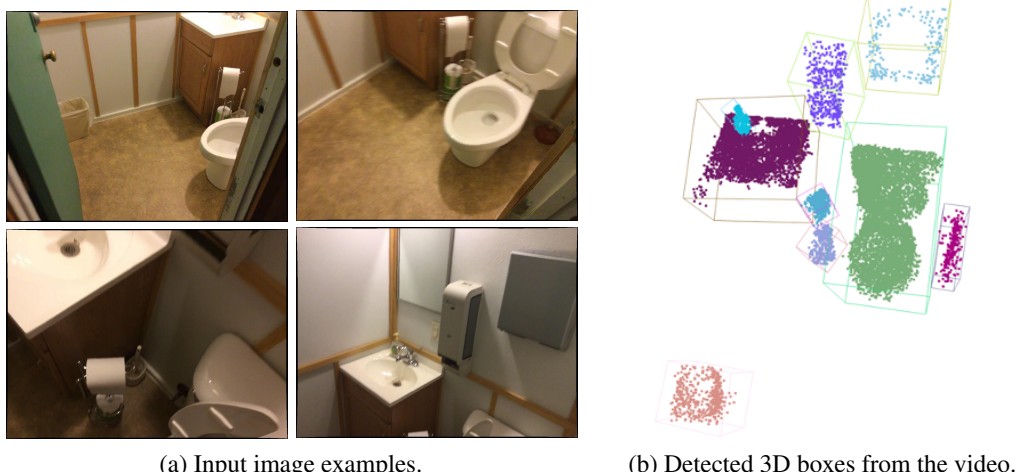

(a) Input image examples.    (b) Detected 3D boxes from the video.

Figure 6: **Open-set 3D object detection from a video sequence.** For the video demos for 3D box generation, please refer to https://ba2det.site.

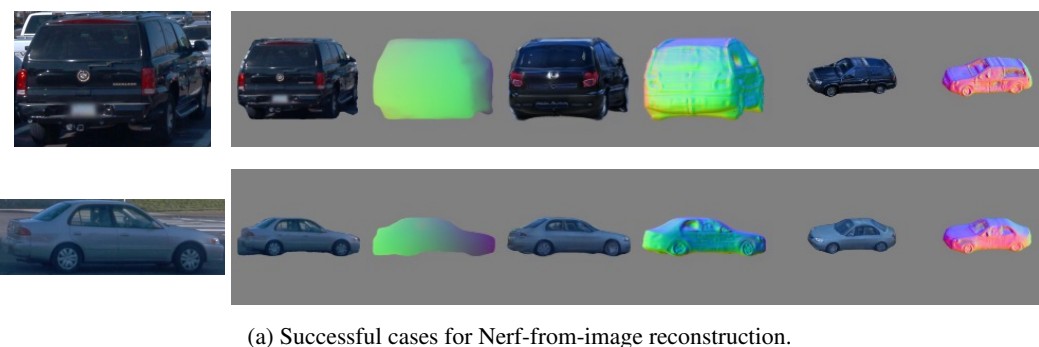

(a) Successful cases for Nerf-from-image reconstruction.

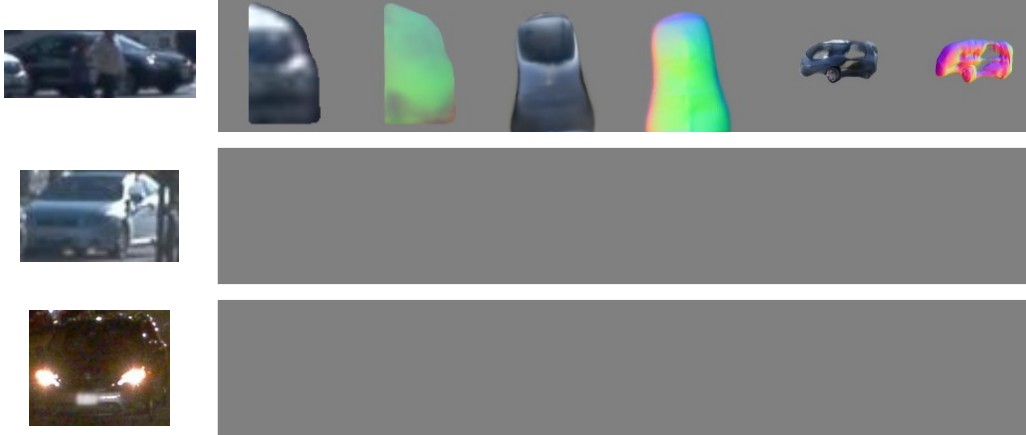

(b) Failure cases for Nerf-from-image reconstruction. The reasons are occlusion, the distant object, at night.

Figure 7: **Qualitative results of Nerf-from-image reconstruction.**

(<10m). However, for WOD, most objects are in [30m,60m]. Please refer to Fig. 3 in the main paper. We find that if the box is less than 80 * 160 pixels, nerf-from-image cannot recognize the object, even if we resize the image to 4x and higher resolution. In WOD, objects less than 80 * 160 pixels are more than 90% of all objects. That means nerf-from-image is an unavailable method on WOD.

- Another problem is occlusion. Nerf-from-image will fail to find the main object in the 2D box.

- Nerf-from-image also fails at night. The object cannot be recognized.

However, our method will not fail in these cases, because our 3D reconstruction is a temporal method. Objects are also easier to be separated in reconstructed point clouds. In summary, we show that even if combined with a 2D detector, nerf-from-image is not a comparable method with our proposed method.

# E    LIMITATIONS AND FUTURE WORK

Although we no longer need 3D annotations in the proposed BA$^2$-Det, it still depends on the 2D annotations. We try to further decrease the dependency on 2D labels and show the ability of open-set 3D object detection in Sec. C.2. However, it is a preliminary exploration. We expect not to use 2D labels anymore finally. And we will continue to work in this direction.

## F  REPRODUCIBILITY STATEMENTS

We will release the training and inference codes to help reproduce our work and the documents will be clearly written. 3D pseudo labeling tools that are based on open-source packages COLMAP (Schonberger & Frahm, 2016)[2] and hloc (Sarlin et al., 2019)[3] will be released. Limited by the license of WOD, the checkpoint of the model trained on WOD cannot be publicly available. However, we will provide it by email if needed. The details of the proposed method BA$^2$-Det, including the implementation details, and network architecture are mentioned in Sec. 4.1, Sec. 4.2, Sec. 4.3, and Sec. 5.2 in the main paper. The data and 2D annotations of WOD[4] are publicly available. The open-set 3D object detection depends on SAM (Kirillov et al., 2023) to generate 2D instance masks, which is also an open-source model.

---

[2]`https://github.com/colmap/colmap`
[3]`https://github.com/cvg/Hierarchical-Localization`
[4]`https://waymo.com/open/`

