# OpenReview forum: "2D-Supervised Monocular 3D Object Detection by Global-to-Local Reconstruction"
_ICLR.cc/2024/Conference — ICLR 2024 Conference Withdrawn Submission_

### Official Review · Reviewer_cbts · 2023-10-22

**Soundness:** 3 good
**Presentation:** 2 fair
**Contribution:** 2 fair
**Rating:** 5
**Confidence:** 3

**Summary:**

This paper presents the BA2-Det method, a novel paradigm for 2D supervised monocular 3D object detection and tracking. The method utilizes global-to-local reconstruction techniques to generate and refine 3D pseudo labels for objects. It achieves comparable performance to fully supervised methods, even with only 2D annotations. BA2-Det demonstrates robustness in different depth ranges and shows potential for detecting open-set 3D objects.

**Strengths:**

1. BA2-Det eliminates the need for expensive 3D ground truth or LiDAR data by utilizing 2D annotations. It introduces a novel pipeline that effectively generates and refines 3D pseudo labels from monocular images, achieving accurate and robust object detection.

2. BA2-Det leverages the power of global and local reconstruction techniques to recover 3D structures from monocular videos. This enables the generation of comprehensive 3D pseudo labels for objects, even in complex and occluded scenes.

3. Experimental results on the Waymo Open Dataset demonstrate that BA2-Det performs on par with fully supervised methods trained with only 10% of the training videos. It also surpasses some pioneering fully supervised methods, indicating its strong performance and potential for various 3D object detection tasks.

**Weaknesses:**

1. Assumptions and Limitations of Monocular 3D Object Detection: The paper assumes that the BA2-Det method is applicable to ordinary videos, but it does not explicitly discuss the limitations of monocular 3D object detection using only 2D annotations. Addressing the inherent limitations and challenges associated with monocular 3D object detection, such as scale ambiguity, occlusion handling, and depth estimation accuracy, would provide a more comprehensive understanding of the method's scope and potential limitations.

2. Simplistic Approach for 3D Object Label Generation: The method for generating 3D pseudo labels relies heavily on global and local reconstructions. While the paper mentions the use of global bundle adjustment (BA) and object-centric local bundle adjustment (LBA), it lacks a comprehensive discussion on the limitations and potential shortcomings of this approach. Additionally, the paper does not provide detailed explanations of the reasoning or justifications behind using specific methods, algorithms, or models within the BA2-Det pipeline.

3. Lack of Comparison with More Recent Methods: The paper only compares BA2-Det with several baselines and a fully supervised BA-Det method. However, it would be beneficial to include a comparison with more recent state-of-the-art methods for a comprehensive evaluation of BA2-Det's performance.

4. Insufficient Analysis of False Positives: The paper briefly mentions the presence of false positives at longer distances but does not provide a detailed analysis of why these false positives occur and how they can be addressed. More insights into the causes and potential solutions would enhance the understanding of the model's limitations.

**Questions:**

The motivation behind this paper is good, but the authors need stronger evidence, including formula derivations and data explanations, to support the effectiveness of their method. Additionally, the presentation of the paper could benefit from further refinement.

---

> ### Author Response · Authors · 2023-11-23
> **Response to Reviewer cbts (Part 1)**
>
> We sincerely thank you for the professional comments. We are inspired and hope our discussion brings more insights.
> ### W1: Assumptions and limitations of monocular 3D object detection
> We use 2D annotation to obtain pseudo 3D labels, and finally train a monocular 3D object detector with these pseudo labels. Similar to other monocular 3D detectors, we also encounter inherent issues such as inaccurate depth estimation and occlusion problems. The main contribution of this paper is proposing the use of weakly supervised methods to train a 3D detector. Compared to using 3D ground truth, our weakly supervised approach has the following limitations.
> - The limitation of reconstructing videos: Our method relies on reconstruction to obtain pseudo labels, so it is limited by the usage conditions of SfM. It is difficult to achieve good reconstruction results when there is no ego-motion or when the camera undergoes severe movement in the dataset. The method doesn't work well for purely static video datasets. However, less than 5% of videos in the driving dataset do not meet the requirements, and we can use only pseudo labels with good reconstruction results to train the model. Therefore, this limitation can be greatly alleviated in driving scenarios. We also apply this method in more general scenes (Fig. 6), where most videos conform to the assumptions of this paper.
> - The limitation of pseudo label quality: (1) Inaccurate orientation: Since many 3D clusters from the reconstructed scene are incomplete due to occlusion and self-occlusion, orientation estimation is more challenging than 3D location and 3D shape estimation. In section 4.2 and 4.3, we try our best to alleviate this problem. However, it still affects the performance and becomes an open problem. (2) Big cluster containing more than one object: When some objects are side-by-side, like in a parking lot, the clustering algorithm may output a big object cluster containing more than one object. We filter this kind of cluster by 3D shape threshold. In the future, we will improve the clustering algorithm to solve this problem. (3) Time-consuming iteratively self-retraining: Although iteratively self-retraining can improve performance a lot, it takes more than double the training time. We will try to decrease the time cost and keep the performance in the future work.
> ### W2: Simplistic approach for 3D object label generation
> The rationality of each module design can be verified in our ablation study (Table 3).  As for why we use specific methods, algorithms, or models, the principle for using them is simple and effective.
> - GBA and LBA are based on SfM, and we implement SfM using the COLMAP library. Implementing GBA and LBA using SfM/COLMAP is because it is a robust and widely-used temporal 3D depth/location estimation method. The main limitations and potential shortcomings using SfM/3D reconstruction are time-consuming and non-ego-motion videos. However, there are many works on 3D reconstruction acceleration [A,B,C]. Besides, non-ego-motion videos are less than 5% in autonomous driving datasets. The negative impact is relatively limited.
> - We use ImmortalTracker as the tracking module, because it is 3D Kalman Filter-based tracker, which is a SOTA 3D tracker without a learnable module.
> - The clustering method is based on the Connected Component (CC) algorithm, a simple graph algorithm implemented by the Scipy library. However, we design the two-stage object clustering method DoubleClustering, much better than just using the CC algorithm.
>
> [A] Ozyesil et al. Robust camera location estimation by convex programming. In CVPR 2015.
>
> [B] Cui et al. HSfM: Hybrid Structure-from-Motion. In CVPR 2017.
>
> [C]  Jin et al. Image matching across wide baselines: From paper to practice. In IJCV 2021.

---

> ### Author Response · Authors · 2023-11-23
> **Response to Reviewer cbts (Part 2)**
>
> ### W3: Comparison with more recent methods
> On the Waymo open dataset (WOD), we have compared BA2-Det with the fully supervised SOTA method BA-Det. On the KITTI dataset, we additionally compare with some recently published methods to show the results more comprehensively. Please refer to the revised paper.
> ### W4: Analysis of false positives
> - False positives at longer distances are mainly caused by the pseudo label distribution. As shown in Figure 3, our pseudo labels can be 0-200m, and so the depth prediction on training set (as labels for retraining) will be in 0-200m. However, in iteratively self-retraining stage, we constrain the depth estimation in 0-75m. That means many too-far objects are predicted near 75m, which are the FP predictions. (However, according to Table 6, 0-75m constraint in retraining is effective for general AP metrics.)
>
> - In this work, we have alleviated this problem by removing objects that are 1 meter away from the farthest predicted distance, i.e., if predicting depth in [0,75m], we filter the object whose depth is more than 74m.
> The improvement is shown in the following table.
>
> |1st iter.|3D AP$_\text{5}$|3D APH$_\text{5}$|LET APL$_\text{50}$|LET AP$_\text{50}$|
> |-|:-:|:-:|:-:|:-:|
> |w/o 74-75m filter|47.46 |35.79|16.54|27.01|
> |w/ 74-75m filter|51.81 |38.66 |17.42 |28.57|
>
> - In the future, we will try the label balancing algorithms to further solve this problem.

---

### Official Review · Reviewer_2jyv · 2023-10-31

**Soundness:** 3 good
**Presentation:** 2 fair
**Contribution:** 2 fair
**Rating:** 5
**Confidence:** 4

**Summary:**

The authors propose a method for learning 3D bounding box detection from 2D bounding box annotations in videos, based on BA-Det. The approach includes three main steps: obtaining point clouds through Structure from Motion (SFM) and clustering, fitting cuboids and refining them by the learning-based refiner to create pseudo-GT of cuboids, and finally, iteratively training the detector and updating pseudo-GT. The proposed method is evaluated on Waymo Open Dataset (WOD) and KITTI, and it performs comparably on KITTI against the fully-supervised baselines. It also shows applicability to indoor scenes with open-set settings combined with segment-anything’s 2D detection.

**Strengths:**

- An extensive ablation study discusses valuable information that significantly affects learning, such as the variation in self-training strategies (Table 5) and the threshold values for the depth of pseudo-GT (Table 6).
- The proposed components of the pipeline add up to the final performance (Table 3)
- Applicability is demonstrated not only for road scenes but also for indoor scenes.

**Weaknesses:**

**Major**
- The paper does not discuss relevant works that learn 3D cuboids from 2D supervision, such as MonoDR [1] and nerf-from-image [2].
- The proposed components of the pipeline are perceived to be a combination of straightforward approaches based on existing techniques without significant technical innovation. For instance, numerous works exist for tracking-based temporal feature integration, clustering employs existing algorithms, and pseudo-GT for training and iterative updates of pseudo-ground truth has also been explored in previous work [3].
- Quantitative evaluation on WOD, which is more challenging than KITTI, shows that the BA-Det and MonoFlex with 10% GT outperform the proposed method significantly by a large margin. I wonder whether this gap becomes closer if 10% GT is used for the proposed method.
- Utilizing a pre-trained extra tracking module in the pipeline and comparing it with the baselines that lack such a module could be seen as an unfair comparison.

**Minor**
- The best results in the table are not highlighted in bold font, making it difficult to discern the superiority of the proposed method.
- The caption of Table 2 and the table header fail to specify the metric used. Ensuring the table is self-contained would aid the reader.


[1] Beker et al. Monocular Differentiable Rendering for Self-Supervised 3D Object Detection. ECCV 2020.

[2] Pavllo et al. Shape, Pose, and Appearance from a Single Image via Bootstrapped Radiance Field Inversion. CVPR 2023.

[3] Zakharov et al. Autolabeling 3D objects with differentiable rendering of SDF shape priors.

**Questions:**

- What is the performance of the proposed method using 10% of GT, compared to BA-Det with the same setting?
- Does this method work comparably with the recent self-supervised pose estimation (+shape reconstruction) method nerf-from-image, when combined with a 2D detector?
- What is the duration of the entire training process?

---

> ### Author Response · Authors · 2023-11-23
> **Response to Reviewer 2jyv (Part 1)**
>
> We really appreciate your valuable comments. We are trying our best to address your concerns and are open to any further discussions.
> ### W1: Discussion about relevant works
> Although MonoDR and nerf-from-image also learn 3D cuboids from 2D supervision, there are significant differences in problem settings and methods between them and us.
> - The core challenge is much different. These methods emphasize the complete object mesh/shape reconstruction, however, we do not reconstruct objects at the mesh level. The reconstruction stage is just for 3D box estimation. The main challenge for us is the object's depth estimation in complex scenes.
> - Besides, they both rely on additional hand-craft class-wise CAD models. However, utilizing video-based 3D reconstruction, we do not need CAD models, as shown in Fig. 6 we can even detect open-set 3D objects.
> - As for depth estimation, they either utilize pretrained depth or depth labels. MonoDR requires a pretrained depth estimator to supervise the depth map for depth estimation. nerf-from-image relies on 2D canonical map labels from synthetic datasets. However, our work does not require these additional data/models/labels.
> - Moreover, since they are fine-grained shape/mesh-based reconstruction methods, an additional pretrained segmentation network is required for obtaining precise object masks. Our model is only supervised with 2D boxes instead of object masks.
> - We also try nerf-from-image in our complex dataset WOD, and the reconstruction performance is not as good as the clean dataset nerf-from-image used. Please refer to the response to Q2 for more details.
>
> ### W2: Technical contributions
> Our pipeline is not just a combination of straightforward approaches based on existing techniques. There are many technical contributions in our work.
> - Although we utilize the simple yet effective Connected Component algorithm in DoubleClustering, the two-stage local-to-global clustering method is totally new. No other work has designed a similar algorithm. The two-stage clustering is also shown effective in our ablation study (Table 12).
> - We propose a novel network $G_\theta$ to learn object 3D size/orientation from the well-reconstructed objects. This network makes it possible to provide more pseudo-labels for occluded objects and these labels play an important role in the following monocular 3D object detector (LBA-Learner) learning stage. (Table 3 & 13)
> - We design novel orientation loss to solve the problem of 180$^\circ$ flipping in pseudo label (Table 14).
> - Our iteratively self-retraining procedure is much different from SDFLabel [3] used on both motivation and method details. What we do in iteratively self-retraining procedure is to evolve and **generalize to the dynamic objects**. SDFLabel uses retraining to sample good pseudo labels by rules to make self-improving. However, we show that in our method, heuristic selection cannot work well (Table 5) and we focus on label refinement.
> - In all, we agree that we also use the pipeline of pseudo label generation, learning from pseudo labels, and retraining in high-level concepts. However, the proposed method is novel and solve the challenging problems in 2D supervised monocular 3D object detection task.
> ### W3&Q1: Using 10% GT
> - We supplement the experiment using the semi-supervised setting with the same 10% GT as BA-Det and MonoFlex.
>
> ||3D AP$_\text{5}$|3D APH$_\text{5}$|3D AP$_\text{50}$|3D APH$_\text{50}$|LET APL$_\text{50}$|LET AP$_\text{50}$|LET APH$_\text{50}$|
> |-|:-:|:-:|:-:|:-:|:-:|:-:|:-:|
> |MonoFlex (10% GT)| 53.68| 52.30 |15.44| 15.22| 28.21| 44.21| 43.23|
> |BA-Det (10% GT)| 57.29 |55.27 |19.70 |19.27| 32.53| 46.91 |45.52|
> |BA$^2$-Det (10% GT)| 65.24|62.95|	25.17	|24.65	|42.55	|58.77	|57.14|
>
>
> With additional 10% GT, our method can outperform BA-Det and MonoFlex.

---

> ### Author Response · Authors · 2023-11-23
> **Response to Reviewer 2jyv (Part 2)**
>
> ### W4: Extra tracking module
> - We use the 3D Kalman Filter-based tracker ImmortalTracker, without any learnable module. So we do not introduce any additional parameters.
> - Besides, in both baseline SfM+BA-Det and fully-supervised SOTA method BA-Det, we also use the same tracking module.
>
> ### W5: Tables
> Sorry for the difficulty readers may have in reading the table. We have fixed it in our revised paper. Thank you again for pointing it out.
>
> ### Q2: Comparing with nerf-from-image
> - We try nerf-from-image in our complex dataset WOD. However, only for object reconstruction, the performance is not as good as it shows in clean datasets CUB Birds and Pascal3D+ Cars. We analyze and discuss why it doesn't work on WOD in the following. For more qualitative results, please refer to section D in the appendix of our revised paper.
> (1) The main problem is for the small objects, i.e., for the objects far from the camera. Since nerf-from-image is designed for object-centric datasets, in which objects are near the camera (<10m). However, for WOD, most objects are in [30m,60m]. Please refer to Fig. 3 in our paper. We find that if the box is less than 80 * 160 pixels, nerf-from-image cannot recognize the object, even if we resize the image to 4x and higher resolution. In WOD, objects less than 80 * 160 pixels are more than 90% of all objects. That means nerf-from-image is an unavailable method on WOD.
> (2) Another problem is occlusion. Nerf-from-image will fail to find the main object in the 2D box.
> (3) Nerf-from-image also fails at night. The object cannot be recognized.
> - However, our method will not fail in these cases, because our 3D reconstruction is a temporal method. Objects are also easier to be separated in reconstructed point clouds.
> - In summary, we show that even if combined with a 2D detector, nerf-from-image is not a comparable method with our proposed method.
>
> ### Q3: Duration of training process
> We require a total of 624 GPU hours for the entire training process. We utilize 8 NVIDIA A40 GPUs to train GBA-Learner for 10 hours, LBA-Learner for 26 hours in the initial iteration, and self-retrain LBA-Learner for 42 hours over two iterations.
> As for fully-supervised BA-Det, the training process is 280 GPU hours.

---

### Official Review · Reviewer_MCGQ · 2023-11-05

**Soundness:** 3 good
**Presentation:** 3 good
**Contribution:** 3 good
**Rating:** 6
**Confidence:** 4

**Summary:**

Amid the rising demand for extensive data in the era of large-scale models, manual annotations pose cost and resource challenges, potentially impeding further progress. Within the domain of monocular 3D object detection, existing research efforts have explored weakly supervised methodologies integrating LiDAR sensors to generate 3D pseudo labels, typically unsuitable for conventional video data. This study introduces an innovative approach, BA2-Det, employing global-to-local 3D reconstruction to supervise the monocular 3D object detector in a purely 2D context.

The proposed paradigm leverages scene-level global reconstruction coupled with global bundle adjustment (BA) to recover 3D structures from monocular videos. To delineate object clusters, the DoubleClustering algorithm is developed. Learning from the complete 3D pseudo boxes generated through global BA, the GBA-Learner predicts 3D pseudo boxes for other occluded objects. Furthermore, training an LBA-Learner with object-centric local BA enables the generalization of 3D pseudo labels to moving objects.

Experimentation on the extensive Waymo Open Dataset reveals that BA2-Det performance aligns with the fully-supervised BA-Det model trained with 10% of videos and even surpasses some pioneering fully-supervised techniques. Additionally, as a pretraining method, BA2-Det demonstrates a remarkable 20% relative improvement on the KITTI dataset. The study also highlights the substantial potential of BA2-Det in detecting open-set 3D objects within complex scenes.

**Strengths:**

(1) The authors present a novel paradigm referred to as BA2-Det, focusing on supervised monocular 3D object detection within a 2D framework. This approach involves generating 3D pseudo labels and facilitating the learning process for the monocular 3D object detector, emphasizing a perspective based on global-to-local 3D reconstruction.
(2) In an effort to overcome the challenge of training a 3D object detector without 3D labels, this paper has developed three key modules: DoubleClustering, GBA-Learner, and LBA-Learner.
(3) In their experiments using various datasets, BA2-Det demonstrates high-quality pseudo labels for training 3D detectors. This approach shows comparable performance to fully supervised methods and a 20% relative improvement on the KITTI dataset when used for pretraining. Additionally, it shows promise for detecting open-set 3D objects in complex scenes.

**Weaknesses:**

(1) In the context of Figure 2, is it possible to train Global BA and Local BA together in an end-to-end manner?
(2) In Table 1, it would be beneficial to compare BA-Det trained with 10% data against BA-Det and MonoFlex, which were trained using similar data proportions. When trained with 100% of the available data, the proposed method shows a performance gap compared to BA-Det. Notably, there exists a 12%, 27%, and 30% gap on 3D AP5, 3D APH5, and 3D AP50, respectively.
(3) In Table3, are there any ablation studies with both 3D and 2D supervision?

**Questions:**

I'm positive about this paper. However I have some concerns in Weaknesses. I hope the authors could response to those questions in the rebuttal. Then I'll make the final decision.

---

> ### Author Response · Authors · 2023-11-23
>
> We are grateful for your valuable comments and constructive advice, which help us a lot to make the paper better.
> ### W1: end-to-end manner
> Thanks for your advice. Leaning in an end-to-end manner has many advantages, which will make the entire method more concise and practical. In our current solution (Figure 2), these two stages are learned separately. The key problem is the clustering algorithm, which is a heuristic algorithm, without the differentiable selection and sampling module.
> However, it is still possible for them to be jointly learned in an end-to-end manner. If the clustering module can be involved in a differentiable network, the end-to-end learning pipeline can be achieved.
> We will aim to make the learning process end-to-end in future work.
> ### W2: 10% data
> - We conduct the additional experiments using only 10% of videos.
>
> ||3D AP$_\text{5}$|3D APH$_\text{5}$|3D AP$_\text{50}$|3D APH$_\text{50}$|LET APL$_\text{50}$|LET AP$_\text{50}$|LET APH$_\text{50}$|
> |-|:-:|:-:|:-:|:-:|:-:|:-:|:-:|
> |MonoFlex (10% data/10% GT)| 53.68| 52.30 |15.44| 15.22| 28.21| 44.21| 43.23|
> |BA-Det (10% data/10% GT)| 57.29 |55.27 |19.70 |19.27| 32.53| 46.91 |45.52|
> |BA$^2$-Det (100% data/0% GT)| 60.01| 44.81 |10.39| 8.98 |22.24| 32.60| 23.86|
> |BA$^2$-Det (10% data/0% GT)|43.47|30.07|1.46|1.19|7.92|11.43|8.95|
>
> It shows that with only 10% videos, the performance will not decrease so much, especially for AP$_\text{5}$ metrics.
> - As for the performance gap when using the same 100% data, this phenomenon is common in most methods using weak annotations. However, we intend to emphasize the comparison between BA$^2$-Det with 100% data and BA-Det with 10% data. With the natural advantage that our method can use more unlabeled data, our method can be comparable to the fully supervised method.
> ### W3: both 3D and 2D supervision
> In Table 3, $N_\theta$ w/ 3D and $N_\theta$ w/ 2D mean 3D and 2D label assignment methods (Sec. 4.3) instead of with 3D and 2D supervision. We use both 2D and 3D supervision for the first row ($N_\theta$ w/ 3D) as well as the second row ($N_\theta$ w/ 2D).
> In detail, that is about prediction and (pseudo) ground truth matching strategy. $N_\theta$ w/ 3D assigns labels using generated 3D pseudo labels. Since we use a CenterNet-based detector, the implementation is supervising 2D/3D classification and regression predictions on the location corresponding to the projected 3D centers on heatmaps. That means some objects without 3D pseudo-labels are not supervised.
> For $N_\theta$ w/ 2D, we supervise 2D/3D classification and regression predictions on the location corresponding to the 2D centers on heatmaps. Those objects without 3D pseudo-labels only have 2D classification and regression loss.
> These two label assignment methods cannot coexist in a network. In other words, we cannot implement both $N_\theta$ with 3D and $N_\theta$ with 2D simultaneously.